# Clinical Performance of Zygomatic Implants—Retrospective Multicenter Study

**DOI:** 10.3390/jcm9020480

**Published:** 2020-02-09

**Authors:** Ruben Davó, Simonas Bankauskas, Remigijus Laurincikas, Ismail Doruk Koçyigit, José Eduardo Mate Sanchez de Val

**Affiliations:** 1Instituto Davó, 03026 Alicante, Spain; jemate@ucam.edu; 2UAB “SB dantuklinika”, LT49452 Kaunas, Lithuania; simonas@clinicdpc.lt (S.B.); remigijuslaurincikas@gmail.com (R.L.); 3Kirikkale University, Dentistry Faculty, Department of Oral & Maxillofacial Surgery, 71450 Kirikkale, Turkey

**Keywords:** edentulous, resorbed maxilla, zygomatic implant

## Abstract

The main objective of this analysis was to evaluate (1) implant survival, (2) biologic complications, and (3) demographics associated with zygomatic implants placed according to the zygomatic anatomy-guided approach (ZAGA). This retrospective multicenter study reviewed data from the charts of 82 consecutive patients who had received 182 zygomatic implants. Patients were fully edentulous (62.2%), partially edentulous (22.0%), or had failing dentition (15.9%). Most patients (87.5%) did not have previous sinusitis and 11.3% had been previously treated for it. Additionally, about half of the patients (53.8%) did not present periodontal pathology, and one-third (36.3%) did, but were subsequently treated. Most implants (93.8%) were loaded immediately, i.e., within 48 h of placement. Implants were followed for 10.5 ± 7.2 months, and all were recorded as surviving and stable at last follow-up. Post-operative complications were infrequent and included sinusitis (10.1%) and peri-implant hyperplasia (0.8%). The low complication rate and 100% implant survival and stability indicate that zygomatic implants offer a viable treatment option when performing graftless restoration of severely resorbed maxilla, including immediate loading protocols.

## 1. Introduction

Rehabilitation with zygomatic implants offers a viable graftless option for treating severely atrophied maxillae [1]. Zygomatic implants are fixtures of 30 to 52.5 mm in length that, when inserted in the body of the malar bone, emerge at the level of the premolars and allow the rehabilitation of the posterior sector without the need to perform bone grafts. The zygomatic implant technique was introduced by the Branemark group who, in their first study, placed 65 zygomatic implants in 27 patients and reported a 100% survival rate after 12 years of follow-up [2]. The most recent systematic review reports a 12 year cumulative survival rate (CSR) of 95.21% based on 68 studies with 4556 zygomatic implants in 2161 patients, and concludes that the technique has high predictability with good clinical results [3].

The placement technique for zygomatic implants has changed over time. Initially, the technique was intrasinusal, i.e., it required opening of a lateral window to the breast wall and elevation of the membrane. To allow a more anatomically and prosthetically driven approach, the original technique has been modified by introducing an extrasinus path for zygomatic implants [4,5,6,7]. Currently, many clinicians modify their zygomatic implant placement technique to fit interpatient anatomic differences in a so-called zygomatic anatomy-guided approach (ZAGA). In this approach, the preparation of the implant site is guided by the anatomy of the area, and no initial window or slot is opened at the lateral wall of the maxillary sinus. Thus, depending on the relationship between the zygomatic buttress and the intra-oral starting point of the zygomatic implant, the path of the implant body will vary from being fully intrasinus to being fully extrasinus, i.e., this approach for the placement of the zygomatic implant is neither ‘internal’ nor ‘external’ to the sinus wall but, instead, promotes the placement of the zygomatic implant according to the anatomy of the patient [8,9].

The main objective of this analysis was to evaluate (1) implant survival, (2) biologic complications, and (3) demographics associated with zygomatic implants placed according to the ZAGA approach.

## 2. Materials and Methods

### 2.1. Study Design

The design of the study was retrospective and included chart review of all patients with severe resorption of the maxilla who had received at least one NobelZygoma 45° implant (Nobel Biocare AB, Göteborg, Sweden); these are newly designed zygomatic implants with a tapered tip—for ease of use at insertion—and a partially unthreaded body that interfaces with the soft tissue and is not necessarily engaged in bone. The study included consecutive patients treated between 20 July 2015 and 11 July 2017 in one of the three participating centers. The participating centers comprised two private clinics, one in Alicante, Spain, and one in Kaunas, Lithuania, as well as a university clinic in Kirikkale, Turkey. The general criteria for inclusion for treatment with zygomatic implants in these clinics were total or partial maxillary edentulism, feasibility of placement of conventional implants in the premaxilla, and impossibility of treating the posterior maxilla either without grafting procedures or with 5 or 6 mm-long implants or pterygoid implants. Conventional posterior implants were used when the implant could emerge at the level of the second premolar. For the quad approach (i.e., in which patients received four zygoma implants each), indications included severe maxillary atrophy, in particular, inadequate bone volume for the placement of even a single dental implant, both anteriorly and posteriorly. The quad zygoma was used as the first option of choice in these patients. Patients with enough maxillary bone to be rehabilitated with conventional implants were excluded, as were patients with acute sinusitis, bruxism, poorly controlled diabetes, or metabolic disorders that may compromise the functionality of the implant. All of the surgery was performed by oral maxillofacial surgeons (R.D., I.D.K.) or general practitioners who had received training in and had prior experience with zygomatic implant placement (S.B.). A separate prosthodontic team performed the prosthetic treatment under the supervision of the treating surgeon. The study included 82 patients with 182 zygomatic implants. The following patient-related data were extracted: age, gender, oral hygiene status (rated, in each case, by the treating clinician as one of four categories: excellent, good, acceptable, or poor), indication (failing dentition, partially edentulous, or fully edentulous), smoking status, presence of pathologic occlusion (defined as class II or class III malocclusion according to the Angle’s classification [10]), history of periodontitis and sinusitis, patient fitness according to the American Society of Anesthesiologists (ASA) status [11], relevant health history including diabetes and tumors or previous irradiation in head/neck area, as well as medication history including use of bisphosphonate therapy. In addition, the following implant, implant site and surgical characteristics were recorded: implant position recorded according to the Fédération Dentaire Internationale (FDI) system, bone quality classified as soft, medium, or hard (based on the Lekholm and Zarb classification [12] simplified to three main categories by merging the middle ones into a single “medium” category), implant number and length, number and type of other conventional implants placed during the surgery, bone grafting prior to implant surgery, bone or soft tissue grafting during implant surgery, final insertion torque, and type of placement depending on the final implant position, i.e., intrasinus, extrasinus, or in the wall of the maxilla.

### 2.2. Surgical Protocol

Subjects were treated according to standard dental care in each participating clinic. Patients diagnosed with periodontitis, in whom the affected teeth were to remain in the mouth, received antibiotic treatment prior to surgery. If the periodontitis-affected teeth were to be removed, the patients were not treated, however, after tooth extraction the sites were allowed to heal for at least 1 month prior to implant placement. Although smoking was not a contraindication for the surgery, all smokers were educated on the possible negative impact of smoking on treatment success, and were strongly encouraged to quit immediately after the surgery. A CT scan or a cone beam computed tomography (CBCT) was performed on all patients prior to the surgery. After the surgical treatment, all patients had an orthopantomogram (OPG) taken, and in cases with complications, additional CT or CBCT scans were taken; these data, however, were not analyzed for the purpose of this study. The surgical area was exposed via an incision in the posterior maxilla and then vertical releasing incisions along the posterior part of the infra-zygomatic crest and anterior to the site of surgery. A small lateral bone window was created to improve visibility of the drilling direction and to enable the dissection of the sinus membrane where needed. Zygomatic implants were then anchored in the zygomatic bone at the level of the maxillary alveolar process and in the zygomatic bone itself as per the standard protocol [13]. All implants were placed according to the individual anatomy of each patient following the ZAGA recommendations [14] and, thus, the resulting position of the implant head was palatal, on the ridge, or vestibular. The simplified placement technique is shown in Figure 1. Post-surgical recommendations included prescription antibiotics (augmentin), anti-inflammatory agents, menthol inhalations, and nasal decongestants. Patients with poor oral hygiene received additional instruction to ensure proper care of implant-supported prosthesis.

### 2.3. Prosthetic Protocol

Immediately after wound closure by suturing, an impression was made using trays, standard impression copings screwed on the abutments, and silicone impression material. The abutments placed at surgery were straight or 17° angulated Multi-unit Abutments (Nobel Biocare AB), Healing Abutments, or Cover Screws (Nobel Biocare AB). Implants were loaded in an immediate protocol (within 48 h after implant insertion), early (between 48 h and 3 months) or delayed (>3 months after implant insertion). Complete arch fixed prostheses with or without metal wire reinforcement were delivered within 24 h of the surgery. After the evaluation of implant stability and soft tissue health 4 to 6 months post-surgery, the patients received their definitive prosthesis. In patients with pathologic occlusion, the prosthetic design included horizontal cantilevers to compensate for malocclusion. The final restoration was fixed and screw-retained.

### 2.4. Outcome Measures

The primary outcome was survival of zygomatic implants. Secondary outcomes included presence of post-operative complications. Post-operative sinusitis was diagnosed by the treating clinician and based on clinical symptoms including difficulty breathing, as well as swelling, redness, or tenderness in the area. Patients with post-operative sinusitis received wide spectrum antibiotics, anti-inflammatory medication, nasal decongestants, and menthol inhalations. No surgical intervention was performed on these patients, i.e., no functional endoscopic nasal surgery (FENS) was performed and no implant was removed. Additional parameters assessed in the study were implant stability and soft tissue health evaluated with the bleeding on probing and modified gingival index score, where score 0 = no bleeding; score 1 = isolated bleeding spots; score 2 = confluent blood; score 3 = profuse bleeding [15].

### 2.5. Statistical Analysis

Patient data were de-identified by the attending clinician, copied from the records into an Excel spreadsheet (Microsoft), and analyzed using SPSS 25 (IBM, Armonk, NY, USA). Descriptive analysis for numeric parameters was performed using means and standard deviations, as well as medians and ranges. Except for patient-level characteristics (gender, oral hygiene status, smoking status, physical status, health history, bisphosphonate therapy, pathologic occlusion, history of periodontitis, history of sinusitis, and indication) the implant was used as the statistical unit of analysis.

## 3. Results

### 3.1. Patient and Implant Baseline Characteristics

Baseline patient and implant characteristics as well as surgical details are listed in Table 1. The mean patient age at surgery was 57 ± 9.4 (range: 33–78). Sixteen patients (19.5%) received 4 zygomatic implants in a quad zygoma technique while three (3.65%) received 3zygomatic implants. Most of the implants (*n* = 96) were placed in the wall of the maxilla and their implant head position was on ridge (*n* = 54), vestibular (*n* = 36) or palatal (*n* = 6); 75 implants were placed extrasinus, with the vestibular (*n* = 61) or on ridge (*n* = 14) implant head position; finally, few (*n* = 11) implants were placed intrasinus, with the resulting vestibular (*n* = 7) or palatal (*n* = 4) head position. Final insertion torque was recorded for 158 (86.8%) of implants and averaged at 40.7 ± 6.8 Ncm (range: 25–55). All but one implant were placed in a one-stage surgery and received straight (*n* = 73) or 17° angulated (*n* = 104) multi-unit abutments on the day of surgery. A cover screw was attached to the remaining single implant and the site was allowed to heal in a two-stage protocol. The vast majority of implants (91.8%) were loaded immediately; for these implants, the mean insertion torque was 41.3 ± 6.8 Ncm (range: 25–55; *n* = 143). Most patients (86.6%) received a provisional prosthesis and 12.2% directly received a final prosthesis. Overall, 53 patients with 110 zygomatic implants were restored with a final prosthesis. The final prostheses were delivered 5.7 ± 4.3 months (range: 0–24) after the surgery. One patient received two separate long-lasting provisional bridges, which were considered as final because of the material (metal-reinforced acrylic) as well as the patient’s economic situation. One patient withdrew after surgery and their prosthetic information was not available; with the exception of prosthetic data, all other data from this patient were included in the statistical analyses.

### 3.2. Clinical Follow-up and Outcome Measures

The mean clinical follow-up was 10.5 ± 7.2 months (range: 0–29 months). All 182 implants were evaluated as stable at the last follow-up visit, resulting in an implant survival rate of 100%. Post-operative complications were assessed at 119 implants (45 patients) and included sinusitis, observed at 12 implants in 5 patients, and hyperplasia around the implant recorded for 1 implant in 1 patient; no fistulas were recorded. Details of implant sites with complications are provided in Table 2. Soft tissue was assessed at only one participating center and included 39 implants in 13 patients. The evaluated sites demonstrated excellent health, with no bleeding on probing and gingival index score of 0 at 31 (79.5%) of implant sites, and gingival index score of 1 (i.e., presence of isolated bleeding spots) at the remaining 8 (20.5%) implant sites in five patients. To better understand the clinical outcomes in the patients suffering from periodontitis, a separate analysis of this patient subset (37 patients, 76 implants) was performed. This subgroup analysis demonstrated a 100% implant survival rate with a mean follow-up of 7.1 ± 0.9 months; post-operative complications in this subgroup were assessed in 36 patients (75 implants) and included sinusitis at 16 implants in 6 patients and hyperplasia around 1implant in 1 patient. In this subgroup, soft tissue assessment was done at 30 implants in 9 patients, and revealed no bleeding on probing and gingival index score of 0 at 24 (80%) of implant sites, and gingival index score of 1 (i.e., presence of isolated bleeding spots) at the remaining 6 (20%) implant sites in 4 patients.

## 4. Discussion

This retrospective analysis evaluated clinical outcomes of treatment with zygomatic implants according to the ZAGA approach. The multicenter setting enabled collection of a large amount of data in a relatively short period of time while avoiding the bias associated with a single operator and a single clinical setting.

The excellent 100% survival rate of the zygomatic implants in this study is consistent with or better than in other studies with a similar clinical follow-up. The recent meta-analysis reports a survival rate of 98.35% for the follow-up of 6 to 12 months [3], while the systematic review on immediately loaded zygomatic implants reports survival rate range of 96%–100% [16]. Another recent meta-analysis, comparing quad zygomas to two zygomatic implants with fixtures from the same manufacturer as the study implant, reports an overall survival rate of 98.0% [17].

Presence of previous sinusitis in 10 of the patients in the study represents 12.2% of the total number of patients, in which 27 implants were placed; only one of them was not treated. Values similar to those were described in the studies by Aparicio et al. [18], Davó et al. [19], Maló et al. [20], and Chow et al. [21]). Out of the 45 patients who were assessed for post-operative sinusitis, 5 experienced the condition, and 3 of these had also previously had sinusitis. Due to these low numbers, as well as the fact that most of the patients with post-operative sinusitis received zygoma implants placed extrasinus (four out of five; Table 2), it is not possible to assess whether prior sinusitis predicts a higher frequency of the condition post-operatively.

Soft tissue assessment in the study revealed that the healing process was uneventful, with no fistulas and with most implants showing no bleeding on probing at the last follow-up. Whether this can be mostly credited to the fact that the part of the implant interacting with the soft tissue is threadless remains to be further investigated, as well as evaluated in a longer follow-up setting.

All zygomatic implants in this study were placed following the ZAGA approach. As a result, the vast majority of the implants were placed extra sinus or in the wall of the maxilla (ZAGA II–IV), and just a minority (6%) in an intrasinus position (Figure 1). This reflects the anatomical situation of the patients presenting maxillary bone atrophy and demonstrates that placement of these implants while avoiding the sinus space is possible in most cases.

Most of the implants (93.8%) included in this study were loaded immediately, i.e., within 48 h of placement. Application of the immediate loading protocol was enabled by the high primary stability, as measured by the final insertion torque, which the authors attribute to the tapered tip of the zygomatic implant used in this study. In addition, patients with severe bone resorption have been treated according to the quad zygoma concept, which enabled the use of immediate loading protocols without prior regenerative approaches. Consequently, most of the patients were provided with a fixed provisional prosthesis, which dramatically shortened time-to-teeth, especially when combined with the fact that no bone grafting was performed prior to or simultaneous with the surgery. The immediate loading of the implants, as performed in this study, is a valuable treatment option because it offers the patient an immediate restoration of function, esthetics, and social confidence [17,22]. In some of the study patients, the provisional acrylic prosthesis served as the final prosthesis, after a realignment at 4 to 6 months after the surgery. This approach of using the provisional prosthesis as the final one has been cited in other studies as likely to be of particular interest in patients with a quad zygoma in whom, due to severe maxillary resorption, the amount of the prosthetic acrylic material tends to be higher and, therefore, the prosthesis is likely to be stronger and longer-lasting.

Treatment of patients with periodontitis is often particularly challenging. In this study, almost half of the patients were diagnosed with the periodontal disease. This number is consistent with a recent Centers for Disease Control and Prevention (CDC) statement, in which about half of Americans aged 30 years or older are estimated to suffer from this disease [23]. Periodontitis has been documented as a risk factor associated with early implant failure as well as peri-implantitis [24,25]. However, the published literature does document successful cases of treatment with zygomatic implants of patients with aggressive periodontitis [26]. In the current study, a subanalysis performed on periodontal patient subgroup demonstrated that the treatment was successful: the patients underwent uneventful healing, all implants survived and remained stable, and patients had healthy peri-implant soft tissue at the last follow-up. It is likely that these positive outcomes are associated with pre-treatment with antibiotics (in those patients in whom the periodontitis-affected teeth remained in the mouth) as well as a minimum of a 1 month healing period (prior to implant placement) in the patients in whom the periodontitis-affected teeth were removed.

Demographic characteristics of the patients treated in this study are similar to those documented in other publications reporting treatment with zygomatic implants. Female patients were more common, as previously reported by Rodriguez-Chessa et al. [27], and most of the patients fell into the age groups of 50–59 and 60–69 years of age. This age distribution is likely to reflect a combination of factors that led to complete edentulism or a complete loss of dental function [3]. Indeed, most of the treated patients (62.2%) were fully edentulous, and only 22.0% and 15.9% respectively were partially edentulous or had failing dentition. The analysis of the presence of an abnormal occlusal relationship was limited by the fact that many patients were fully edentulous; however, among the 67 assessed patients, only 13.4% were diagnosed with pathologic occlusion. Regarding oral hygiene status, one-third of the patients were classified as having poor oral hygiene, while the periodontal assessment revealed that half of the patients had a history of periodontitis. These results are not surprising given the dentition status of the treated patients, and are in agreement with previously published studies [28]. However, it is important that the assessment of the oral hygiene status in this study was not standardized, i.e., it was based on a retrospective chart review and followed the guidelines individual to each clinic.

The main limitations of this study are associated with the short follow-up of the patients. In addition, some of the outcomes were not assessed in all of the participating centers. Nevertheless, the inclusion of consecutive patients from three separate clinics provides an important insight into the real-world performance of zygomatic implants in treatment of edentulous maxillae.

Within the limitations of this clinical study, rehabilitation with zygomatic implants resulted in a low complication rate and a 100% implant survival rate, confirming it as a viable treatment option when performing graftless restoration of severely resorbed maxilla. The analysis of the demographic characteristics indicated that most patients were female, between 50 and 69 years of age, fully edentulous, and 50% with a history of periodontitis.

## Figures and Tables

**Figure 1 jcm-09-00480-f001:**
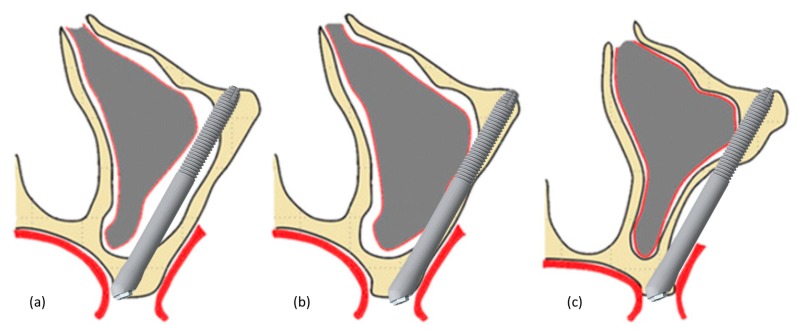
Illustration of the zygomatic implant placement techniques: intrasinus (**a**), in the wall of the maxilla (**b**), and extrasinus (**c**).

**Table 1 jcm-09-00480-t001:** Baseline characteristics.

Characteristics		*n* (%)	*n* Evaluated
Patients			82
Gender	Female	53 (64.6%)	82
	Male	29 (35.4%)	
Oral hygiene status	Excellent	1 (1.2%)	82
	Good	15 (18.3%)	
	Acceptable	36 (43.9%)	
	Poor	30 (36.6%)	
Smoking status	Yes	16 (19.5%)	82
	No	66 (80.5%)	
Patient physical status	ASAI	48 (71.6%)	67
	ASAII	18 (26.9%)	
	ASAIII,	1 (1.5%)	
	ASAIV, V, VI	0	
Health history	Diabetes	0	82
	Cancer	0	
	Rheumatoid arthritis	2 (2.4%)	
Bisphosphonate therapy	Yes (over 1 year)	1 (1.2%)	82
	No	81 (98.8%)	
Pathologic occlusion	Yes	9 (13.4%)	67
	No	58 (86.6%)	
History of periodontitis	No	43 (53.8%)	80
	Yes, treated	29 (36.3%)	
	Yes, not treated	8 (10.0%)	
History of sinusitis	No	70 (87.5%)	80
	Yes, treated	9 (11.3%)	
	Yes, not treated	1 (1.3%)	
Indication	Failing dentition	13 (15.9%)	82
	Partially edentulous	18 (22.0%)	
	Fully edentulous	51 (62.2%)	
Implants/implant sites			182
Position	Central incisor	3 (1.6%)	182
	Lateral incisor	23 (12.6%)	
	Canine	8 (4.4%)	
	1^st^ premolar	4 (2.2%)	
	2^nd^ premolar	80 (44.0%)	
	1^st^ molar	63 (34.6%)	
	2^nd^ molar	1 (0.5%)	
Implant length (in mm)	30.0	2 (1.1%)	182
	32.5	8 (4.4%)	
	35.0	31 (17.0%)	
	37.5	7 (3.8%)	
	40.0	29 (15.9%)	
	42.5	40 (22.0%)	
	45.0	31 (17.0%)	
	47.5	13 (7.1%)	
	50.0	10 (5.5%)	
	52.5	11 (6.0%)	
Bone quality	Soft	19 (10.4%)	182
	Medium	129 (70.9%)	
	Hard	34 (18.7%)	
Bone grafting prior to implant surgery	Yes	0	182
	No	182 (100%)	
Bone grafting at implant surgery	Yes	0	84
	No	84 (100%)	
Soft tissue grafting at implants surgery	Yes	39 (21.4%)	182
	No	143 (78.6%)	
Loading protocol	Immediate	167 (93.8%)	178
	Early	9 (5.1%)	
	Delayed	2 (1.1%)	

**Table 2 jcm-09-00480-t002:** Characteristics of implant sites with post-surgical complications.

Nr	Gender	Indication	Oral Hygiene	Smoking	History of Sinusitis	History of Perio-Dontitis	Bone Quality	Soft Tissue Grafting during Implant Surgery	Insertion Torque	Placement Technique Body & Collar	Position of Implant Head	Loading Protocol	Provi-Sional Prosthesis
Complication: post-operative sinusitis													
1	male	failing dentition	good	no	yes, treated	no	medium	no	40	extrasinus	vestibular	immediate	yes
2							medium	no	40	extrasinus	vestibular		
3	male	fully edentulous	acceptable	no	no	no	medium	no	40	extrasinus	vestibular	immediate	yes
4							medium	no	35	maxillary wall	vestibular		
5							medium	no	40	maxillary wall	vestibular		
6	male	fully edentulous	acceptable	no	yes, treated	no	medium	no	40	extrasinus	vestibular	immediate	yes
7							medium	no	40	extrasinus	vestibular		
8							medium	no	40	extrasinus	vestibular		
9							medium	no	40	extrasinus	vestibular		
10	female	fully edentulous	good	yes	no	yes, not treated	medium	no	50	extrasinus	vestibular	immediate	yes
11	female	failing dentition	good	no	yes, treated	yes, treated	hard	yes	not reported	intrasinus	palatal	immediate	yes
12							medium	yes	45	maxillary wall	on ridge		
Complication: hyperplasia around zygomatic implant													
1	male	fully edentulous	poor	yes	no	yes, treated	hard	yes	40	maxillary wall	on ridge	immediate	yes

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
