# Peer review of "Clinical Performance of Zygomatic Implants—Retrospective Multicenter Study"

_jcm, 2020, doi:10.3390/jcm9020480_

Round 1
Reviewer 1 Report
Dear Authors,
Thank you for all changes.
Reviewer 2 Report
Article presents interesting issue. It is generally well written.
Minor English editing is demanded. Please remove yellow marker form the text.
Tittle
I suggest changing it to: Clinical performance of zygomatic implants - retrospective multicentre study.
Abstract
The aim should be exactly the same as in the main text.
The same applies to conclusions.
Main text
Introduction
Purpose
The main objective of this analysis was to evaluate (1) implant survival, (2) biologic complications and (3) demographics associated with zygomatic implants placed according to the ZAGA approach based on real world evidence collected from consecutive patients treated in three different clinics.
I suggest shortening the aim, for example:
The main objective of this analysis was to evaluate (1) implant survival, (2) biologic complications and (3) demographics associated with zygomatic implants placed according to the ZAGA approach.
Materials and methods
In this section only information on materials and methods used should be mentioned. Other data should be either deleted or moved elsewhere.
Information on clinics taking part in the study should be mentioned in this section. Please also elaborate shortly on the qualifications of the dentist preforming surgery and prosthetic treatment.
Line 56 This study followed the STROBE guidelines
The sentence is redundant.
Please specify inclusion and exclusion criteria.
Was informed consent from patients obtained? Did you obtain the consent of Ethical Committee?
Was x-ray diagnostics performed before / after surgery?
How long was the follow-up ?
Surgical and prosthetic protocols should be written as separate sections. Both protocols should be described in detail. Now there is no data on prosthodontic protocol.
Did patients received any post-surgery recommendations?
Line 138
What do you mean by subgroup (analysis)?
Line 139 (range: 0 – 29)
What do you mean by range? Percent?
How the sinusitis was treated?
Results
Please elaborate in method section on what is meant by good, acceptable, poor oral hygiene. Please address this issue in the discussion.
Please elaborate in method section on placement technique (table 2). Please address this issue in the discussion.
Please elaborate in method section on provisional and final prosthesis. Please address this issue in the discussion.
Discussion
Lines 172-174
Of note, in the clinical experience of the authors, predictable achievement of high primary stability would not have been possible with the predecessor of the study implant.
This should be supported by articles.
Lines 175-177
Consequently, most of the patients were provided with a fixed provisional prosthesis, which dramatically shortened time-to-teeth, especially when combined with the fact that no bone grafting was performed prior to or simultaneous with the surgery.
This sentence introduces new information on prosthodontic procedure. Please add appropriate statement in materials and methods section.
Please discuss results obtained by other authors.
Lin 180 CDC statement
Please define CDC.
Line 205 some of the outcomes were not assessed in all of the participating centers.
What do you mean? It should be mentioned in Method section.
Conclusions
Please write conclusions as points as they were written in objective section. Address all points/issues that are in objective section.
Author Response
Response to Reviewer 2 Comments
Article presents interesting issue. It is generally well written.
Response. We thank the reviewer for their general comment as well as the careful review.
Minor English editing is demanded. Please remove yellow marker form the text.
Response. The yellow marker was introduced to highlight the changes applied during the first revision round. In this revision, the changes are again highlighted. Once approved, the highlighting will be removed.
Tittle. I suggest changing it to: Clinical performance of zygomatic implants - retrospective multicentre study.
Response. We have changed the title according to the reviewer’s suggestion.
Abstract
The aim should be exactly the same as in the main text.
Response. We have re-written the aim in the abstract to match the one listed in the main body of the manuscript:
“The main objective of this analysis was to evaluate (1) implant survival, (2) biologic complications and (3) demographics associated with zygomatic implants placed according to the ZAGA approach.”
The same applies to conclusions.
Response. We have re-written the aim in the conclusions to match the one listed in the main body of the manuscript.
“Within the limitations of this clinical study, rehabilitation with zygomatic implants resulted in a low complication rate and a 100% implant survival rate confirming it as a viable treatment option when performing graftless restoration of severely resorbed maxilla. The analysis of the demographic characteristics indicated that most patients were female, between 50 and 69 years of age, fully edentulous, and 50% with a history of periodontitis.”
Main text
Introduction
Purpose
The main objective of this analysis was to evaluate (1) implant survival, (2) biologic complications and (3) demographics associated with zygomatic implants placed according to the ZAGA approach based on real world evidence collected from consecutive patients treated in three different clinics.
I suggest shortening the aim, for example:
The main objective of this analysis was to evaluate (1) implant survival, (2) biologic complications and (3) demographics associated with zygomatic implants placed according to the ZAGA approach.
Response. We shortened the aim as per reviewer’s suggestion.
Materials and methods
In this section only information on materials and methods used should be mentioned. Other data should be either deleted or moved elsewhere.
Response. Most of the additional information in the materials and methods had been added during the first round of reviews (please see below for details). Although resulting in a rather lengthy materials and methods section, we hope that this information may be of use to the general readership of the journal. We therefore would like to keep this extended version of this section.
1st round of reviews, reviewer 1:
Point 7. How would you define pathologic occlusion (table1)?
Response 7. Pathologic occlusion was defined as class II and class III occlusal relationship according to the Angle’s classification (ANGLE 1899). Malocclusion was always taken into account when planning the implant treatment by compensating with the use of horizontal cantilevers. This information has now been added to the manuscript.
Point 8. In table 1 some information appear twice (gender, oral hygiene status, ect). Why?
Response 8. This mistake has appeared during the automatic formatting process at submission. We apologize for the confusion. The table has been corrected.
Point 11. How post-operative sinusitis was assessed? Who did this?
Response 11. Due to lack of a consensus statement on the definition of sinusitis, post-operative sinusitis was diagnosed according to each clinic and by the treating clinician, and was based on clinical symptoms including difficulty breathing as well as swelling and tenderness around the area.
1st round of reviews, reviewer 2:
Point 4. It could be worth considering to rewrite this section in more detail include some more information about Oral Hygiene Status, what does it mean that hygiene was acceptable and what was the difference between poor and good oral hygiene?
Response 4. Oral hygiene status was classified based on presence of plaque and bleeding on probing around all dentition in the patient’s mouth as well as the state of the patient’s denture if applicable. Patients with poor oral hygiene received additional instruction to ensure proper care of implant-supported prosthesis. This information has been added to the manuscript.
Point 5. Please write more about periodontal assessment, classification and smoking status: describe in more detail smoking levels (how many cigarettes per day? how long?) Pathologic occlusion- what this means and what has been done to eliminate it? Was evaluated Satisfaction Grade Assessment, can you elaborate more this subject based on recent literature? It could be important for your discussion.
Response 5. Periodontal assessment has been performed by the treating clinician and based on clinical symptoms. If patients presented with periodontitis around teeth that were to remain in their mouths after the surgery, antibiotic treatment was initiated prior to implant placement. If the periodontitis-affected teeth were to be removed, no treatment was initiated; however, these patients were excluded from immediate placement protocols. Smoking status was assessed but no patient was denied treatment based on how many cigarettes per day they smoked. All smokers were educated on the possible negative effect of smoking on treatment success and were strongly encouraged to quit, at least immediately after the surgery, and many followed the recommendation. Pathologic occlusion was evaluated based on Angle’s classification and defined as class II and III occlusal relationship. During treatment planning, horizontal cantilevers in the prosthetic restorations were used to compensate for malocclusion. Patient satisfaction was not assessed in the study. The information regarding periodontitis, smoking, and malocclusion has been added to the manuscript.
Information on clinics taking part in the study should be mentioned in this section.
Response. The information on the participating clinics has been added to the materials and methods section:
“The participating centers comprised two private clinics, one in Alicante, Spain, and one in Kaunas, Lithuania, as well as a university clinic in Kirikkale, Turkey.”
Please also elaborate shortly on the qualifications of the dentist preforming surgery and prosthetic treatment.
Response. All of the surgeries were performed by maxillofacial surgeons or general practitioners who had received training in and had prior experience with zygomatic implant placement. A separate prosthodontic team performed the prosthetic treatment under the supervision of the treating surgeon. This information has been added to the manuscript.
“All of the surgeries were performed by oral maxillofacial surgeons (R.D., I.D.K.) or general practitioners who had received training in and had prior experience with zygomatic implant placement (S.B.). A separate prosthodontic team performed the prosthetic treatment under the supervision of the treating surgeon.”
Line 56 This study followed the STROBE guidelines. The sentence is redundant.
Response: The sentence was deleted.
Please specify inclusion and exclusion criteria.
Response. The general inclusion and exclusion criteria have been added to the materials and methods section:
“The general inclusion for treatment with zygomatic implants in these clinics were total or partial maxillary edentulism, feasibility of placement of conventional implants in the premaxilla, and impossibility to treat the posterior maxilla without grafting procedures or with 5 or 6 mm-long implants, or pterygoid implants. Conventional posterior implants were used when the implant could emerge at the level of the second premolar. For the quad approach (i.e., in which patients received 4 zygoma implants each) indications included severe maxillary atrophy, in particular, inadequate bone volume for the placement of even a single dental implant both anteriorly and posteriorly. The quad zygoma was used as the first option of choice in these patients. Patients with enough maxillary bone to be rehabilitated with conventional implants were excluded as were patients with acute sinusitis, bruxism, poorly controlled diabetes, or metabolic disorders that may compromise the functionality of the implant.”
Was informed consent from patients obtained? Did you obtain the consent of Ethical Committee?
Response. The study was performed as a retrospective chart review, in which patient data were deidentified for the purpose of the analysis. Therefore, no Ethical Committee approval was needed nor applied for. All patients were informed of the surgery and had given their consent to the procedure.
Was x-ray diagnostics performed before / after surgery?
Response. A CT scan or a CBCT were performed in all patients prior to the surgery. After the surgical treatment all patients had an OPG taken, and cases with complications additional CT or CBCT were also taken. These data, however, were not included in the study. This information has been added to the materials and methods section:
“A CT scan or a CBCT were performed in all patients prior to the surgery. After the surgical treatment all patients had an OPG taken, and cases with complications additional CT or CBCT were also taken; these data however were not analysed for the purpose of this study.”
How long was the follow-up ?
Response. The mean clinical follow-up was 10.5 ± 7.2 months (range: 0 – 29 months). This information is present in the result section (subsection 3.2, Clinical follow-up and outcome measures).
Surgical and prosthetic protocols should be written as separate sections. Both protocols should be described in detail. Now there is no data on prosthodontic protocol.
Response. The surgical and prosthetic protocols have been separated into two subsections and more detailed information on the prosthetic protocol has been added to the materials and methods section.
Materials and Methods
“Subsection 2.2; Surgical protocol
The surgical area was exposed via an incision in the posterior maxilla and then vertical releasing incisions along the posterior part of the infra-zygomatic crest and anterior to the site of surgery. A small lateral bone window was created to improve visibility of the drilling direction and to enable the dissection of the sinus membrane where needed. Zygomatic implants were then anchored in the zygomatic bone at the level of the maxillary alveolar process and in the zygomatic bone itself as per the standard protocol [14].”
“Subsection 2.3; Prosthetic protocol
Immediately after wound closure by suturing, an impression was made using trays, standard impression copings screwed on the abutments and silicone impression material. The abutments placed at surgery were straight or 17° angulated Multi-unit Abutments (Nobel Biocare AB), Healing Abutments or Cover Screws (Nobel Biocare AB). Implants were loaded in an immediate (within 48 hours after implant insertion), early (between 48 hours and 3 months) or delayed (> 3 months after implant insertion) protocol. Complete arch fixed prostheses with or without metal wire reinforcement were delivered within 24 hours of the surgery. After the evaluation of implant stability and soft tissue health 4 to 6 months post-surgery, the patients received their definitive prosthesis. In patients with pathologic occlusion, the prosthetic design included horizontal cantilevers to compensate for malocclusion. The final restoration was fixed and screw-retained.”
Did patients receive any post-surgery recommendations?
Response. All patients were prescribed antibiotics (augmentin), anti-inflammatory agents, menthol inhalations and nasal decongestants. This information is added to the materials and methods section:
“Post-surgical recommendations included prescription antibiotics (augmentin), anti-inflammatory agents, menthol inhalations and nasal decongestants.”
Line 138 What do you mean by subgroup (analysis)?
Response. The subgroup analysis included all patients who were diagnosed with periodontitis and was performed and added to the manuscript based on the 1st round of revisions. This is now reworded in the results section for better clarity.
“To better understand the clinical outcomes in the patients suffering from periodontitis, a separate analysis of this patient subset (37 patients, 76 implants) was performed. This subgroup analysis demonstrated a 100% implant survival rate with the mean follow-up of 7.1 ±0.9 months; post-operative complications in this subgroup were assessed in 36 patients (75 implants) and included sinusitis at 16 implants in 6 patients and hyperplasia around 1 implant in 1 patient. In this subgroup, soft tissue assessment was done at 30 implants in 9 patients, and revealed no bleeding on probing and gingival index score of 0 at 24 (80%) of implant sites, and gingival index of 1 (i.e., presence of isolated bleeding spots) at the remaining 6 (20%) implant sites in 4 patients.”
Line 139 (range: 0 – 29) What do you mean by range? Percent?
Response. The range was given in months. We apologize for this omission. The sentence was corrected:
“The mean clinical follow-up was 10.5 ± 7.2 months (range: 0 – 29 months).”
How the sinusitis was treated?
Response. Patients with post-operative sinusitis received wide spectrum antibiotics, anti-inflammatory medication, nasal decongestants and menthol inhalations. No surgical intervention was performed in these patients, i.e., no functional endoscopic nasal surgery (FENS) was performed and no implant was removed. This information has been added to the materials and methods section.
“Patients with post-operative sinusitis received wide spectrum antibiotics, anti-inflammatory medication, nasal decongestants and menthol inhalations. No surgical intervention was performed in these patients, i.e., no functional endoscopic nasal surgery (FENS) was performed and no implant was removed.”
Results
Please elaborate in method section on what is meant by good, acceptable, poor oral hygiene. Please address this issue in the discussion.
Response. Oral hygiene status was rated by the treating clinician; specifically, each clinician could choose one of four ratings from a drop-down menu: excellent, good, acceptable, or poor. These ratings followed the individual practice guidelines, however, in general were based on plaque, bleeding, and overall hygiene status of the prosthesis where applicable. This is now clarified in the methods section, and discussed in the discussion.
Materials and methods
“The following patient-related data were extracted: age, gender, oral hygiene status (rated in each case by the treating clinician as one of four categories: excellent, good, acceptable, or poor), indication (failing dentition, partially edentulous, or fully edentulous), smoking status, presence of pathologic occlusion (defined as class II or class III malocclusion according to the Angle’s classification [11]), history of periodontitis and sinusitis, patient fitness according to the ASA status [12], relevant health history including diabetes and tumours or previous irradiation in head/neck area as well as medication history including use of bisphosphonate therapy.”
Discussion
“Regarding the oral hygiene status, one-third of the patients were classified as having poor oral hygiene, while the periodontal assessment revealed that half of the patients had a history of periodontitis. These results are not surprising, given the dentition status of the treated patients, and are in agreement with previously published studies [28]. However, it is important that the assessment of the oral hygiene status in this study was not standardized, i.e., it was based on a retrospective chart review and followed the guidelines individual to each clinic.”
Please elaborate in method section on placement technique (table 2). Please address this issue in the discussion.
Response. The placement technique (simplified ZAGA classification) is now illustrated in Figure 1. A short paragraph has also been added to discuss the placement technique in the discussion section.
“All zygomatic implants in this study were placed following the ZAGA approach. As a result, the vast majority of the implants were placed extra sinus or in the wall of the maxilla (ZAGA II-IV) and just a minority (6%) in an intrasinus position (Figure 1). This reflects the anatomical situation of the patients presenting maxillary bone atrophy and demonstrates that placement of these implants while avoiding the sinus space is possible in most of the cases.”
Please elaborate in method section on provisional and final prosthesis. Please address this issue in the discussion.
Response. The details concerning the provisional and final prosthesis have been added to the materials and methods as well as discussion section.
Materials and methods, subsection 2.3, Prosthetic Protocol
“Immediately after wound closure by suturing, an impression was made using trays, standard impression copings screwed on the abutments and silicone impression material. The abutments placed at surgery were straight or 17° angulated Multi-unit Abutments (Nobel Biocare AB), Healing Abutments or Cover Screws (Nobel Biocare AB). Implants were loaded in an immediate (within 48 hours after implant insertion), early (between 48 hours and 3 months) or delayed (> 3 months after implant insertion) protocol. Complete arch fixed prostheses with or without metal wire reinforcement were delivered within 24 hours of the surgery. After the evaluation of implant stability and soft tissue health 4 to 6 months post-surgery, the patients received their definitive prosthesis. In patients with pathologic occlusion, the prosthetic design included horizontal cantilevers to compensate for malocclusion. The final restoration was fixed and screw-retained.”
Discussion
“Most of the implants (93.8%) included in this study were loaded immediately, i.e., within 48 hours of placement. Application of the immediate loading protocol was enabled by the high primary stability, as measured by the final insertion torque, which the authors attribute to the tapered tip of the zygomatic implant used in this study. In addition, patients with severe bone resorption have been treated according to the quad zygoma concept, which enabled the use of immediate loading protocols without prior regenerative approaches. Consequently, most of the patients were provided with a fixed provisional prosthesis, which dramatically shortened time-to-teeth, especially when combined with the fact that no bone grafting was performed prior to or simultaneous with the surgery. The immediate loading of the implants such as performed in this study is a valuable treatment option because it offers the patient an immediate restoration of function, esthetics, and social confidence [18,23]. In some of the study patients, the provisional acrylic prosthesis served as the final prosthesis, after a realignment 4 to 6 months after the surgery. This approach of using the provisional prosthesis as the final one has been cited in other studies is likely to be of particular interest in patients with a quad zygoma in whom due to severe maxillary resorption the amount of the prosthetic acrylic material tends to be higher and therefore the prosthesis is likely to be stronger and longer-lasting.”
Discussion
Lines 172-174 Of note, in the clinical experience of the authors, predictable achievement of high primary stability would not have been possible with the predecessor of the study implant.
This should be supported by articles.
Response. Unfortunately, most publications do not document zygomatic implant stability. The sentence was therefore deleted to exclude the comparison to previous experience and is now only based on the results obtained in the current study. The information on the implant geometry designed to facilitate high primary stability has been added to the materials and methods section.
“The design of the study was retrospective and included chart review of all patients with severe resorption of the maxilla who received at least one NobelZygoma 45° implant (Nobel Biocare AB, Göteborg, Sweden), newly designed zygomatic implants with a tapered tip – for ease of use at insertion - and a partially unthreaded body that interfaces with the soft tissue and is not necessarily engaged in bone.”
Lines 175-177 Consequently, most of the patients were provided with a fixed provisional prosthesis, which dramatically shortened time-to-teeth, especially when combined with the fact that no bone grafting was performed prior to or simultaneous with the surgery.
This sentence introduces new information on prosthodontic procedure. Please add appropriate statement in materials and methods section.
Please discuss results obtained by other authors.
Response. More detailed information on the prosthetic protocol has been added to the materials and methods section, and the prosthetic protocol has now been discussed in the discussion section.
“Subsection 2.3; Prosthetic protocol
Immediately after wound closure by suturing, an impression was made using trays, standard impression copings screwed on the abutments and silicone impression material. The abutments placed at surgery were straight or 17° angulated Multi-unit Abutments (Nobel Biocare AB), Healing Abutments or Cover Screws (Nobel Biocare AB). Implants were loaded in an immediate (within 48 hours after implant insertion), early (between 48 hours and 3 months) or delayed (> 3 months after implant insertion) protocol. Complete arch fixed prostheses with or without metal wire reinforcement were delivered within 24 hours of the surgery. After the evaluation of implant stability and soft tissue health 4 to 6 months post-surgery, the patients received their definitive prosthesis. In patients with pathologic occlusion, the prosthetic design included horizontal cantilevers to compensate for malocclusion. The final restoration was fixed and screw-retained.”
Discussion
“Most of the implants (93.8%) included in this study were loaded immediately, i.e., within 48 hours of placement. Application of the immediate loading protocol was enabled by the high primary stability, as measured by the final insertion torque, which the authors attribute to the tapered tip of the zygomatic implant used in this study. In addition, patients with severe bone resorption have been treated according to the quad zygoma concept, which enabled the use of immediate loading protocols without prior regenerative approaches. Consequently, most of the patients were provided with a fixed provisional prosthesis, which dramatically shortened time-to-teeth, especially when combined with the fact that no bone grafting was performed prior to or simultaneous with the surgery. The immediate loading of the implants such as performed in this study is a valuable treatment option because it offers the patient an immediate restoration of function, esthetics, and social confidence [18,23]. In some of the study patients, the provisional acrylic prosthesis served as the final prosthesis, after a realignment 4 to 6 months after the surgery. This approach of using the provisional prosthesis as the final one has been cited in other studies is likely to be of particular interest in patients with a quad zygoma in whom due to severe maxillary resorption the amount of the prosthetic acrylic material tends to be higher and therefore the prosthesis is likely to be stronger and longer-lasting.”
Lin 180 CDC statement Please define CDC.
Response. The acronym has been now defined:
“This number is consistent with a recent Centers for Disease Control and Prevention (CDC) statement, in which about half of 30 year old or older Americans are estimated to suffer from this disease [22].”
Line 205 some of the outcomes were not assessed in all of the participating centers.
What do you mean? It should be mentioned in Method section.
Response. The data in this study were based on a retrospective chart review and not all of the participating centers have recorded some of the outcomes. The outcomes that were not assessed at all of the centers included soft tissue health evaluated using the bleeding on probing and gingival index score. This information has been clarified in the results section.
“Soft tissue was assessed at only one participating center and included 39 implants in 13 patients. The evaluated sites demonstrated excellent health, with no bleeding on probing and gingival index score of 0 at 31 (79.5%) of implant sites, and gingival index of 1 (i.e., presence of isolated bleeding spots) at the remaining 8 (20.5%) implant sites in 5 patients.”
Conclusions
Please write conclusions as points as they were written in objective section. Address all points/issues that are in objective section.
Response. The conclusions have been re-written as per reviewer’s request:
“Within the limitations of this clinical study, rehabilitation with zygomatic implants resulted in a low complication rate and a 100% implant survival rate confirming it as a viable treatment option when performing graftless restoration of severely resorbed maxilla. The analysis of the demographic characteristics indicated that most patients were female, between 50 and 69 years of age, fully edentulous, and 50% with a history of periodontitis.”

Round 2
Reviewer 2 Report
Thanks for all the changes.